# How emotional changes affect skin odor and its impact on others

**Masako Katsuyama**[1]*, **Tomomi Narita**[1], **Masaya Nakashima**[1], **Kentaro Kusaba**[1], **Masatoshi Ochiai**[1], **Naomi Kunizawa**[1], **Akihiro Kawaraya**[2], **Yukari Kuwahara**[2], **Masahiro Horiuchi**[3], **Koji Nakamoto**[3]

1 Shiseido Global Innovation Center, Shiseido Co. Ltd., Yokohama, Kanagawa, Japan, 2 Corporate Research & Development Division, Takasago International Corporation, Hiratsuka, Kanagawa, Japan, 3 Takata Koryo Co., Ltd. Amagasaki City, Hyogo, Japan

* masako.katsuyama@shiseido.com

**Data Availability Statement:** All relevant data are within the paper and its Supporting Information files.

**Funding:** The authors received no specific funding for this work.

## Abstract

The gas emanating from human skin is known to vary depending on one's physical condition and diet. Thus, skin gas has been gaining substantial scholarly attention as an effective non-invasive biomarker for understanding different physical conditions. This study focuses on the relationship between psychological stress and skin gas, which has remained unclear to date. It has been deduced that when participants were subjected to interviews confirmed as stressful by physiological indicators, their skin emitted an odor similar to stir-fried leeks containing allyl mercaptan and dimethyl trisulfide. This characteristic, recognizable odor appeared reproducibly during the stress-inducing situations. Furthermore, the study deduced that individuals who perceive this stress odor experience subjective tension, confusion, and fatigue (Profile of Mood States scale). Thus, the study findings indicate the possibility of human nonverbal communication through odor, which could enhance our understanding of human interaction.

## Introduction

Human emotions can be conveyed in gestures, vocal pitch, facial expressions, and other forms, and people can often perceive and understand these nonverbal indicators. The study focuses on the effects of body odor as a form of nonverbal communication.

Scholars have reported two main routes for the generation of body odor. The first is degeneration by oxidation that occurs through the interaction between microorganisms and sebum/sweat on the skin's surface [1,2]. The second is odor generated through the skin from inside the body. This type of odor is described as transcutaneous blood volatile organic compounds or skin gas [3–5]. For a long time, scholars have recognized that body odor can change along with changes in physical condition, such as during illness. Recent studies have proposed that skin gas can be analyzed to determine the state of the body along with other biological indicators, such as saliva, urine, and blood [6]. Since then, research in this field has advanced significantly.

**Competing interests:** The authors have declared that no competing interests exist.

For example, in the medical field, it has been shown that diabetic patients have a higher acetone content in their skin gas than healthy individuals [7,8]. In addition to this finding, interventions have been pursued to manage various states of the disease. Similarly, many studies reported that dogs can detect the presence of early-stage cancer in their owners through odor. As a result, dogs are being trained for cancer detection [9,10]. Alternatively, we investigated the relationship between skin gas and the effects of diet, physical condition, and aging in healthy volunteers instead of people suffering from illnesses or disorders. The results indicated that older people emit 2-nonenal [11], and people with constipation emit high levels of p-cresol [12].

Based on these findings, skin gas is deemed a beneficial means of investigating the body's condition, including diseases and general physical health. The study further hypothesized that psychological changes may alter skin gas and ultimately impact others. Therefore, the study investigated changes in the odor of skin gas during stress-inducing sessions, with the objective of identifying key components, and ultimately, confirming the impact on others.

Our findings indicate that humans reproducibly emit discernible odors through the skin containing allyl mercaptan and dimethyl trisulfide during interviews that induce psychological tension. Furthermore, we found that when other people smell this odor, they undergo subjective tension, confusion, and tiredness. According to the results of these studies, skin gas may contain ingredients that convey human psychological conditions.

## Materials and methods

### Effects of tension on skin gases

**Subject.** The study subjects consisted of 40 healthy Japanese women aged 35–44 years. As the sampling skin gas in this study was performed on the hands, it was necessary to avoid tobacco odor contaminants found on the hands of smokers; hence, the target population was nonsmokers. In addition, as body odor is known to differ between men and women and with age, the study was initially restricted to women. The subjects underwent skin-gas sampling as well as saliva sampling and electrocardiography.

In another odor assessment study conducted to confirm that the applicability of this phenomenon transcended the female Japanese subjects to include four healthy, nonsmoking Japanese men in their 20s and 30s, three Chinese men and women, and two French men. The subjects were told about the purpose, methods, anticipated clinical benefits, and disadvantages of the study before it started. All participants provided written informed consent, and the study complied with the ethical principles established by the Declaration of Helsinki. Furthermore, the study received the approval of the Shiseido Ethics Committee (Approval Number: C01402, C01555, C01577, C01824).

Participants in the study were instructed (a) not to consume odorous food one day before the study, (b) avoid contact with any strongly smelling objects, and lastly, (c) not to wear any fragrance.

**Test methods and skin-gas collection methods.** Exam outline: In a room kept under a constant temperature and humidity, the test was conducted twice on different days: the first time with the task of answering questions from an unknown interviewer for 20 min and the second time on a day of relaxing in a chair and reading a magazine. On day one, subjects were fitted with an ECG monitor before the interview, and their electrocardiographic activity was measured continuously. Their nondominant hand was kept in a skin-gas sampling bag during the interview and while they were reading a magazine. Their saliva was collected three times. The first collection was taken as a baseline while resting at home during the same time frame

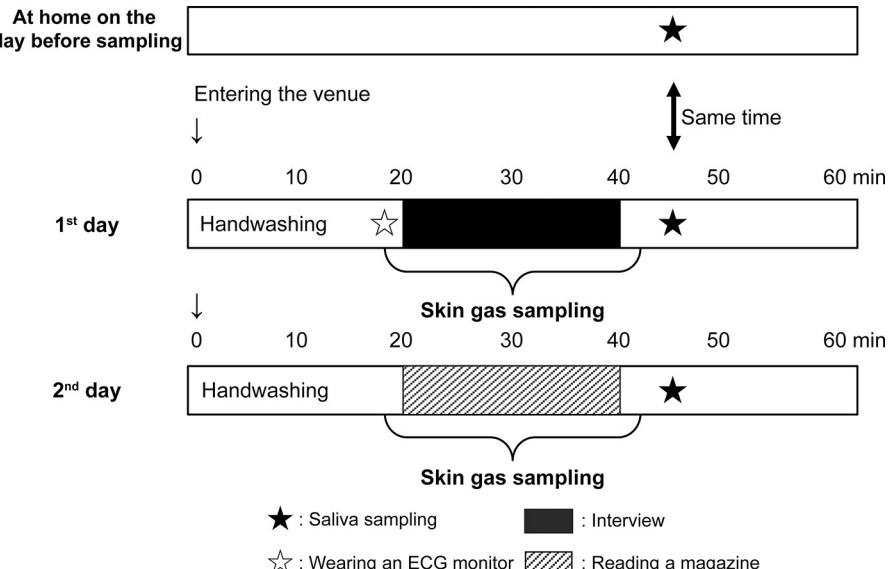

**Fig 1. Effects of tension on skin gases.** Skin-gas samples were collected from each subject in two situations: under stress (answering questions for 20 min from an interviewer whom they did not know) and at rest (sitting on a chair and relaxing while reading a magazine on a different day). Subjects washed their hands with unscented soap before sampling. The electrocardiographic activity was measured continuously by an ECG monitor (Ledar Circ, Sumitomo Dainippon Pharmaceutical). Saliva samples were collected three times. The first measurement refers to samples collected as a baseline while the subjects were at home resting during the same time frame as that of the interview but on a different day. The second sample was collected after the interview, and the third one was collected after the subjects had read a magazine.

as of the interview but on a different day. The second collection was taken after the interview, and the third one was after the subjects read a magazine (Fig 1).

Under the skin-gas sampling procedure, participants washed their hands with unscented soap and their nondominant hand was covered with a sampling bag made of polyvinyl fluoride resin film (Tedlar®) with a one-way stopcock attached. First, the stopcock was opened, and excess air was pushed out of the collection bag in which the hand was placed. Next, a bag (500 mL) filled with nitrogen gas was connected via a short silicone tube, and the nitrogen-filled bag was pressed to move the nitrogen into the collection bag. The empty nitrogen bag was then removed from the stopcock. After a certain amount of time (Fig 2), a storage bag (ANA-LYTIC-BARRIER™, GL Sciences Corporation) was connected via a short silicone tube to the stopcock. The collected gas was transferred from the collection bag to the storage bag by pressing the collection bag. This gas was used as the sample.

**Physiological index measurements.** *Autonomic status*. Four electrodes were placed under the left and right subclavian bones and the tenth left and right ribs. The electrocardiographic activity was measured continuously by an electrocardiogram monitor (Ledar Circ, Sumitomo Dainippon Pharmaceutical).

Based on the electrocardiogram data, a power spectrum analysis was done on the heartbeat "R wave" intervals using autonomic nervous system activity analysis software (Sumitomo Dainippon Pharmaceutical, FLUCLET®). The intensities of the high frequency (HF) (ranging from 0.2 to 2 Hz) and low frequency (LF) (ranging from 0.04 to 0.15 Hz) components were determined. Additionally, the LF/HF component was used as an activity index of the sympathetic nervous system.

*Salivary cortisol measurements*. Their saliva was collected three times. The first collection was done as a baseline while the subjects were resting at home during the same time frame as

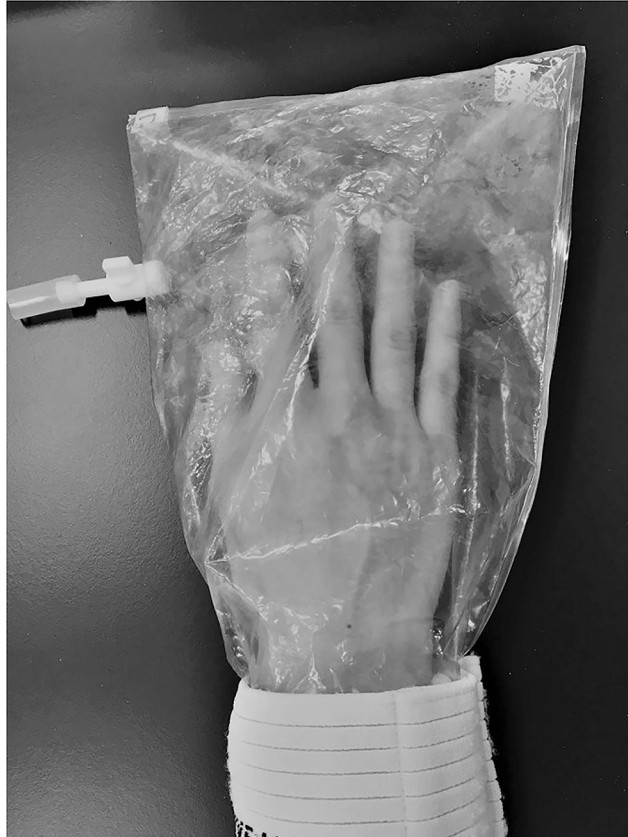

**Fig 2. Collection bag for skin gas emanating from subjects' hands.** Hands washed with unscented soap were covered with sampling bags made from a fluorinated ethylene-propylene copolymer material. The air inside was replaced with nitrogen and recovered after a certain period.

that of the interview but on a different day. The second collection was done after the interview, and the third one was done after the subjects read a magazine. We observed this process to avoid cortisol changes resulting from circadian rhythms, which are particularly variable in the morning. Salivary cortisol was quantified using ELISA (Cortisol Salivary Immunoassay Kit Salimetrics LLC).

*Odor assessment*. A panel of four odor experts sniffed and recorded the characteristics of skin gases collected during stress and at rest respectively. The samples were labeled by number and the examiners made their assessments without knowing the context in which the samples were taken. A five-point Likert-type scale (0 = Not smelly at all, 1 = Slightly smelly, 2 = Smelly, 3 = Distinctly smelly, 4 = Very strongly smelly) evaluated the intensity of the characteristic smell. The results revealed that all gas samples taken during the interview had a characteristic odor, like that of stir-fried leek, while samples taken at baseline did not have any perceivable odor. The skin-gas samples taken during a tense state were subjected to the following analysis for component identification of the odor resembling stir-fried leek.

**Identification of characteristic odor components.**   In order to identify the main components of the characteristic odor from the gas samples collected during a tense moment found by the sensory evaluation, a component analysis was performed in the following order.

**1) Analysis 1**
Preparation of standard sample

Dimethyl trisulfide (Sigma-Aldrich) was diluted to 0.05% (w/v) with dichloromethane.

**Samples**

A total of 1 L of sampled gas (taken during the interviews) was collected for analysis. Several gas samples were integrated by aspirating them to a 6 L canister, the inner surface of which had been deactivated and prepared with negative pressure.

**Gas sampling from canister to Cooled Injection System CIS 4**

The 6 L canister (ENTECH INSTRUMENTS, INC., Simi Valley, CA, USA) was connected to Cooled Injection System CIS 4 (GERSTEL GmbH & Co. KG, Mulheim an der Ruhr, Germany) using a carrier gas line T-connected with an inert tube. CIS4 with a glass-wool liner was cooled to −150˚C during sampling; a 165 mL calculated gas sample was trapped from a canister.

Gas chromatography-mass spectrometry/olfactometry (GC-MS/O) analysis

The GC6890N/5973 inert system (Agilent Technologies, Santa Clara, CA, USA) was equipped with CIS4 (GERSTEL) and an olfactory detection port ODP3 (GERSTEL). The odor was identified by one odor expert and one GC analyst who had learned the odor characteristics from the odor expert. The GC-MS/O was fitted with a DB-WAX column (60 m x 0.25 mm i.d., 0.25 μm of film thickness, Agilent J & W). The column temperature program was set as follows: the initial temperature was kept at 40˚C for 5 min and then increased to 200˚C by 5˚C/min. Helium was used as the carrier gas at a constant linear velocity of 32 cm/s. The CIS4 temperature was programmed as follows: the initial temperature was kept at −150˚C for 0.01 min and then increased to 220˚C by 12˚C/s. The ion source temperature was set at 230˚C. Injections were conducted in a solvent vent mode. The ion source was operated at 70 eV using electron ionization (EI), and the mass spectrometer was set in SCAN mode.

**2) Analysis 2**

**Preparation of standard gas samples**

Allyl mercaptan (Wako Pure Chemical Industries, Ltd., Osaka, Japan) was diluted with iso-propyl myristate (Nikko Chemical Co., Ltd., Tokyo, Japan) to a concentration of 1 ppm and 5 ppm. 1 μL of each concentration was put into a 1 L sampling bag (Smart Bag PA, GL Sciences Inc.) filled with 400 mL of nitrogen and used as standard gas samples after equilibrating for over six hours at room temperature before the experiment. The final concentrations of the standard gas samples prepared from each of the 1 and 5 ppm solutions were ca. 2.5 ng/L and ca. 12.5 ng/L, respectively.

**Dynamic headspace sampling**

Each sampling bag was connected to an air sampling pump JAS-15M II (Japan Analytical Industry Co., Ltd., Tokyo, Japan) with a silicone tube. An adsorption cartridge mini-PAT (Japan Analytical Industry Co., Ltd.), composed of a sample tube filled with Tenax GR wrapped around foil, was connected between the bag and the air pump. By sucking 400 mL of the gas at 200 mL/min using the pump, odorants in each sample were collected onto the adsorbent. The adsorbed odorants were desorbed at 280˚C by curie point injector JCI-55 (Japan Analytical Industry Co., Ltd.) and then subjected to GC-O analysis.

**Solid-phase microextraction (SPME) sampling**

An SPME fiber coated with 85 μm carboxen/polydimethylsiloxane (CAR-PDMS) (Supelco Inc., Bellefonte, PA, USA) was used. The fiber was exposed to each sample gas for 30 min at room temperature. The adsorbed odorants were then analyzed using GC-MS.

**Gas chromatography-olfactometry (GC-O) analysis**

A GC-2010 plus (Shimadzu Co., Kyoto, Japan) was equipped with an olfactory detection port ODP275 (GL Sciences Inc., Tokyo, Japan). The odor was confirmed by two odor experts and a GC analyst. The GC was fitted with a DB-Sulfur SCD column (60 m x 0.32 mm i.d., 4.2 μm of film thickness, Agilent J & W). The column temperature program was set as follows:

the initial temperature was kept at 35˚C for 3 min and then increased to 250˚C by 10˚C/min. Helium was used as the carrier gas at a constant linear velocity of 31 cm/s. The injector temperature was set at 230˚C. Injections were conducted in a splitless mode.

**Gas Chromatography-mass spectrometry (GC-MS) analysis**

The GC-MS analyses were performed using an Agilent GC7890A/5975C system (Agilent Technologies, Santa Clara, CA, USA) equipped with a DB-Sulfur SCD column (60 m x 0.32 mm i.d., 4.2 μm of film thickness, Agilent J & W). The column temperature program was set as follows: the initial temperature was kept at 35˚C for 3 min and then increased to 250˚C by 10˚C/min. Helium was used as the carrier gas at a constant linear velocity of 31 cm/s. The injector and the ion source temperature were set at 230˚C. Injections were conducted in a splitless mode. The ion source was operated at 70 eV using EI, and the mass spectrometer was set in the selected ion monitoring (SIM) mode. The mass to be analyzed for allyl mercaptan was $m/z$74.

**Subjective reaction when smelling the model odor.** The study developed a model odor using the components identified in the previous section and examined their psychological impact on other people.

**Samples of model odor**

Allyl mercaptan (AM) and dimethyl trisulfide (DMTS) were diluted with isopropyl alcohol to create five levels of concentration (AM: DMTS = [0.5: 2.5 ppb], [1: 5 ppb], [2: 10 ppb], [5: 25 ppb], [10: 50 ppb]). 10 μL of each concentration was applied to a small piece of cotton and the subjects sniffed it.

**Subjects for subjective reaction test**

The subjects consisted of 33 Japanese women aged 35–44 years. All participants were informed of the purpose, methods, anticipated clinical benefits, and disadvantages of the study before it started. We obtained written informed consent from all participants, and the study complied with ethical principles established by the Declaration of Helsinki. Furthermore, the study received the approval of the Shiseido Ethics Committee (Approval Number: C01555, C01577). To minimize the effects of food and other odors on the subjective effects of smell, the subjects were instructed (a) not to consume odorous food one day before the study and (b) avoid contact with any strongly smelling objects. They were also instructed not to wear any fragrance.

**Assessing the subjective state**

The study first confirmed the per-subject threshold before assessing their subjective states. A model tension-stress odor whose concentration was adjusted to the odor threshold of each subject was used as the test sample. The subjective state was assessed before and after sniffing the odor using the Profile of Mood States (short version, POMS 2), 2nd edition. POMS 2 contains 35 questions to assess the following moods: anger–hostility (AH), confusion–bewilderment (CB), depression–dejection (DD), fatigue–inertia (FI), tension–anxiety (TA), vigor–activity (VA), and friendliness (F). POMS 2 is a self-reporting questionnaire for rapid assessment of short-term or sustained emotional states, and widely used not only for patients but also for healthy subjects. The higher the T-scores for AH, CB, DD, FI, and TA, the more negative the mood. Conversely, positive mood states, such as VA and F, indicate that the higher the T score, the better the mood [13–15].

Before taking measurements, participants were instructed to relax and stay comfortably seated for at least 10 min. They then filled out the POMS 2. Next, each participant inhaled their threshold concentration of the tension-stress odor applied to a cotton and placed under their nose for 2 min. Immediately following this, the participants repeated the POMS 2 test. POMS was answered on a 5-point Likert-style scale and calculated as a T score using standard POMS 2 calculation methods for seven mood status assessments.

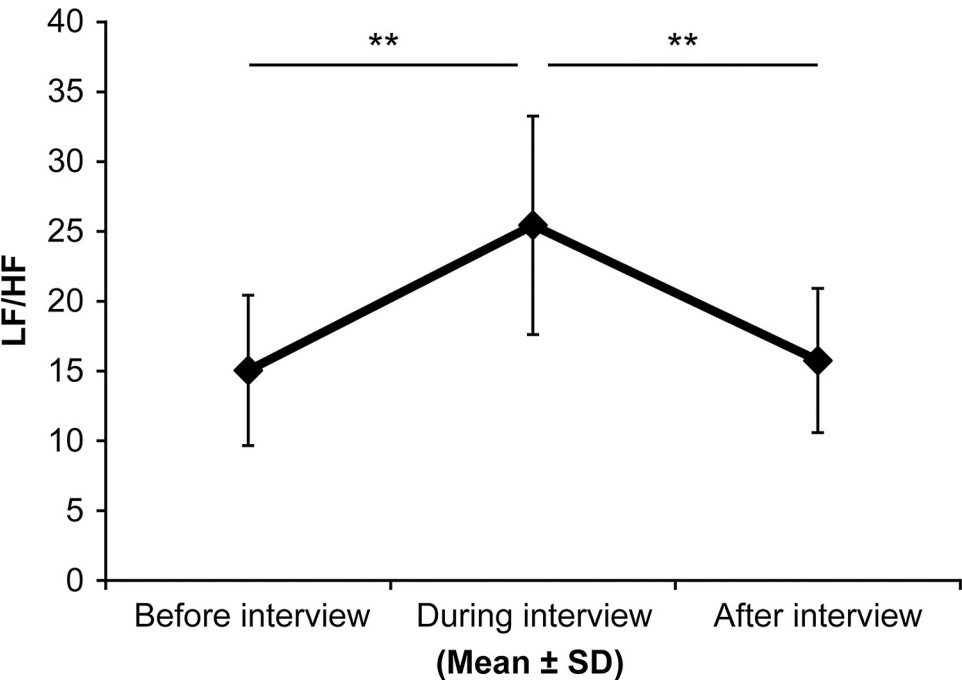

**Fig 3. Autonomic nervous system activity during a stress-inducing interview.** The R wave intervals for the heart rate were calculated using electrocardiogram data. The high frequency intensity levels (HF, 0.2 to 2 Hz range) and low frequency (LF, 0.04 to 0.15 Hz range) components were identified. The LF/HF components were calculated using the activity index of the sympathetic nervous system. The LF/HF values collected during the interview were significantly higher than that of the previous and subsequent measurements, which indicates that the sympathetic nervous system was dominant (**$p < 0.01$ after Bonferroni correction).

## Statistical analysis

The results of the test samples and each experiment are presented as means ± SDs. Bonferroni's method was used for the significance tests for the sympathetic nervous system activity and for the results of the salivary cortisol levels. The relationship between odor intensity and autonomic nervous activity was assessed using pairwise nonparametric correlations (Spearman's). Finally, Wilcoxon's signed-rank test determined significant differences in the results of the psychological index. A p-value <0.05 on both sides was considered statistically significant. All statistical analyses were performed with EZR (Saitama Medical Center, Jichi Medical University, Saitama, Japan, version 1.54), a graphical user interface for R (The R Foundation for Statistical Computing, Vienna, Austria).

## Results

### Physiological responses under emotional tension

To examine the changes in skin-gas odor resulting from psychological distress, the study investigated physiological response changes in the components of skin gas gathered from the subjects' hands during the interview sessions that purposefully induced stress. For one physiological response, the heart rate "R wave" intervals were calculated using electrocardiogram data. The HF intensity levels (0.2 to 2 Hz range) and LF (0.04 to 0.15 Hz range) components were identified. The LF/HF components were calculated using the activity index of the sympathetic nervous system.

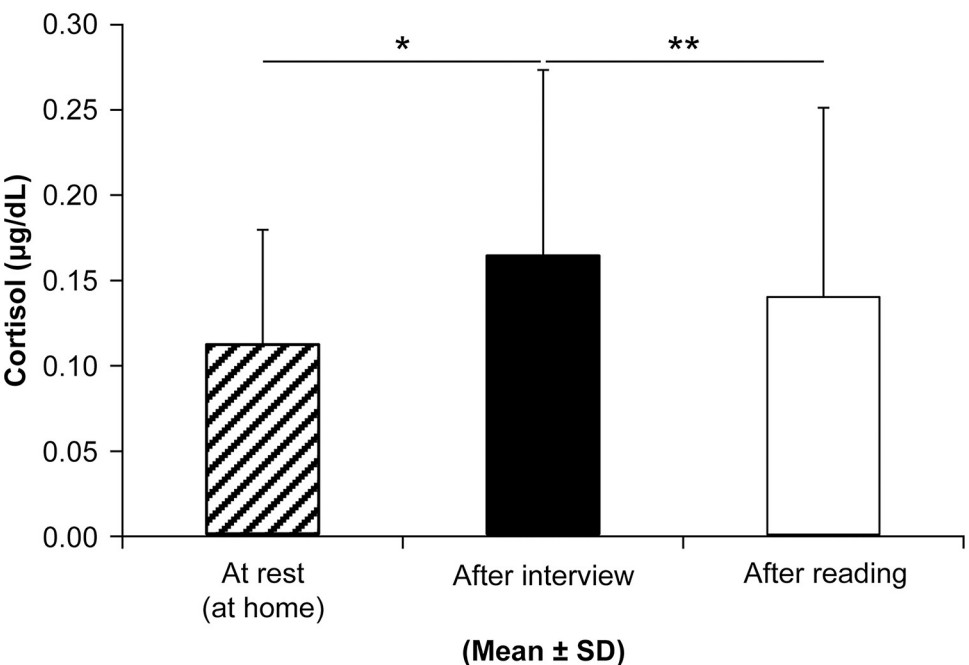

**Fig 4. Salivary cortisol measurements.** Saliva was collected three times. The first sample was collected within the same time frame as that of the interview but on a different day while the subjects were at home resting. The second sample was collected after the interview, and the third one was collected after the subjects had read a magazine. Salivary cortisol levels after the interview were significantly higher than those at baseline ($^*$p < 0.05) and those while the subjects were reading ($^{**}$p < 0.01, after Bonferroni correction).

The LF/HF during the interview was significantly higher than those of previous and subsequent measurements, which indicates that the sympathetic nervous system was dominant ($^{**}$p < 0.01 after Bonferroni correction) (Fig 3).

Salivary cortisol levels after the interview were significantly higher than those at baseline were ($^*$p < 0.05), and while subjects were reading ($^{**}$p < 0.01, after Bonferroni correction) (Fig 4).

### Sensory assessment of skin-gas samples gathered under emotional tension

Four expert odor raters sniffed the skin-gas samples taken during stress and at rest, respectively. When they found a perceived odor, they rated and recorded its characteristics and intensity on a 5-point scale ranging as 0 = Not smelly at all, 1 = Slightly smelly, 2 = Smelly, 3 = Distinctly smelly, 4 = Very strongly smelly. Samples were numbered and evaluated by the rater without prior knowledge of the conditions under which the sample was collected. Thus, four odor experts recorded the presence of a unique odor similar to stir-fried leeks common to multiple samples, all of which had been collected at the time of the stress-inducing interviews.

Additionally, the intensity of this characteristic stir-fried leek-like odor was positively correlated with the rate of increase in sympathetic nerve activity during the interview (Spearman's correlation coefficient, r = 0.66; p <0.01; Fig 5). To confirm that this phenomenon was not limited to the female Japanese subjects; various other participants (Japanese: four males, Chinese: one female, two males, French: two males) were tested while they experienced mental tension. Thus, the same characteristic odor was detected from all participants while they were experiencing mental tension. On the other hand, this stir-fried leek-like odor was not detected at all in samples taken during cycling runs, conducted to cause an increase in heart rate due to physical exercise instead of psychological factors (Fig 6).

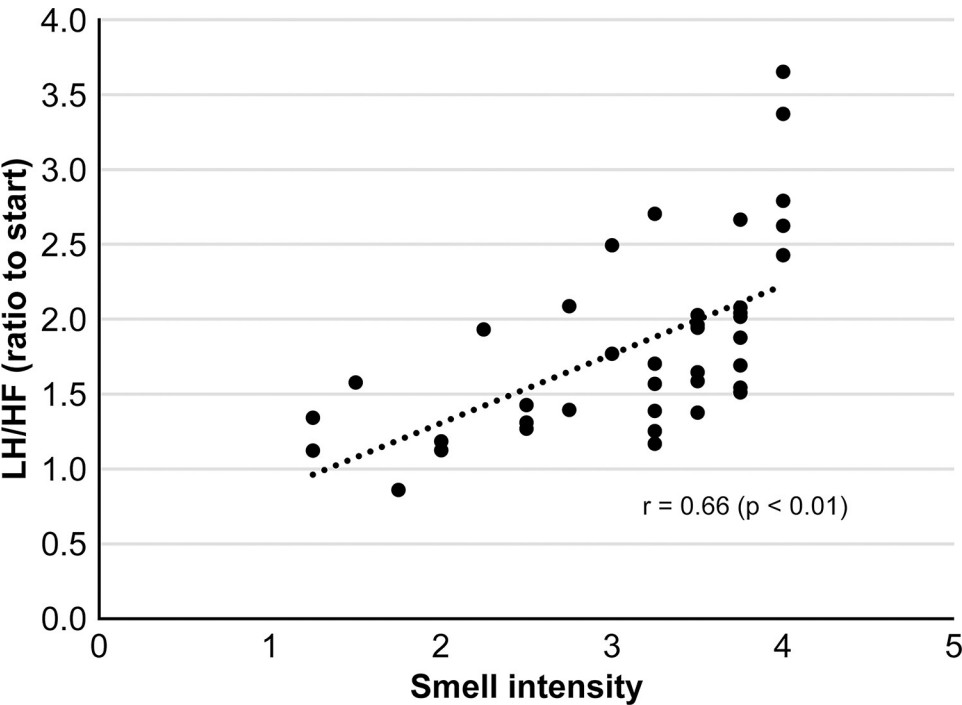

**Fig 5. Relationship between the intensity of tension-stress odor and autonomic nervous activity during the interview.** A panel of four odor experts evaluated the collected skin-gas samples to compare differences between those emanated during stress and at baseline. From all interview samples, we discovered a characteristic odor resembling stir-fried leek. The intensity of this characteristic odor was positively correlated with the rate of increase in sympathetic nerve activity during the interview compared with before the interview (Spearman's correlation coefficient: r = 0.66 ($p < 0.01$)).

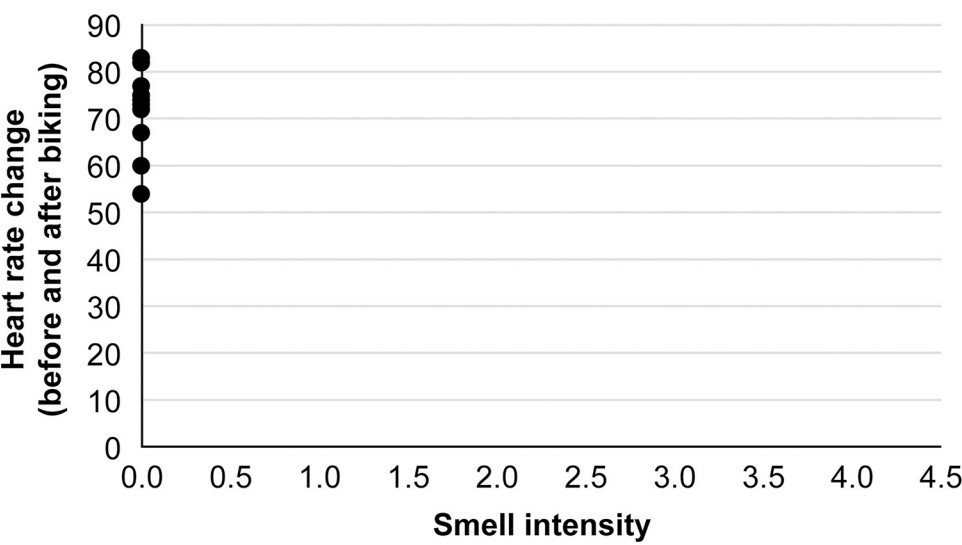

**Fig 6. Relationship between the intensity of tension-stress odor during exercise and changes in heart rate.** A panel of four scent experts evaluated the collected skin-gas samples to identify odor differences among individuals exercising and at rest. No characteristic odor resembling stir-fried leek was observed in any samples when the heart rate increased because of exercise.

## Identifying characteristic odorants

GC-O analysis investigated the main components contributing to the characteristic stir-fried leek-like odor of skin gas collected during the stress interviews. The results revealed that there were a few odor-active components similar to the scent.

**1) Characteristic component 1.** A sample collected during the stress interview showed a peak close to the characteristic odor with a retention time (RT) = 20.571 minutes. (Fig 7). Moreover, the standard DMTS was detected at RT = 20.571 min. (Fig 7A and 7D). At the peak of RT = 20.571 min., these ions were detected by extracting mass chromatograms with m/s = 126 and 79 as a feature of DMTS (Fig 7B and 7C). Hexanol and nonanal were detected before and after the peak at RT = 20.571. From the correlation ($R2 = 0.9984$) between the RTs and the retention index (RI) of the three components registered in the MS private library database, DMTS was identified as the target substance (Fig 8). Furthermore, using GC-MS/O analysis, we determined that the peak of the RT = 20.571 is DMTS from the quality of its odor.

**2) Characteristic component 2.** One of the few odor-active components similar to the tension-stress odor had a more intense sulfurous and alliaceous scent than the others. From its odor description and GC RT, we estimated it to be AM. To confirm the presence of AM in the skin gas, GC-MS (SIM) analyses were performed. On SIM chromatograms of the skin gas, a peak in each skin gas sample was detected at the same RT as AM, and the peak intensity of the skin gas was higher than that of 2.5 ng/L and lower than that of 12.5 ng/L (Fig 9).

In subsequent GC-O analyses, RT and odor quality of the characteristic odor detected in skin gas were the same as those in AM. The odor intensity of the skin gas was slightly stronger than the 2.5 ng/L standard sample was, which was consistent with their peak intensity in GC-MS analysis. From these results, the target component was identified as AM.

## Subjective effects of the tension odor

The model tension odor, formulated according to the threshold, was continuously sniffed by participants for 2 min. POMS 2 measured the participants' subjective status at baseline and immediately after sniffing the sample.

The study observed that negative scores of "tension–anxiety" ($p < 0.01$), "confusion–bewilderment" ($p < 0.01$), and "fatigue–inertia" ($p < 0.01$) significantly increased after sniffing the model tension odor (Wilcoxon's signed-rank test) (Fig 10).

## Discussion

Scholars recently noted that humans release a wide variety of gases from the skin, whose components may act as a barometer for indicating the state of the body. For example, diabetics characteristically release acetone, whereas those with liver disease typically emit ammonia. Moreover, the study confirmed that the human skin releases specific components due to changes in the body unrelated to illnesses, such as aging and constipation. Alternatively, studies have reported body odor in samples collected mainly from the armpit, which is caused by various emotional changes. The majority of studies have examined the reactions of humans or dogs after smelling different states of human odors [16–18]. Reports of human odors estimated by statistical methods from gas chromatography data do not indicate how such odors affect human emotions [19]. Therefore, this study hypothesized that changes in the psychological state chemically affect skin gas and its components affect human psychology. Our purpose was to identify the main components of skin gas released during certain emotional states and examine their psychological effects. Among the various psychological changes present, this study chose to use the Trier social stress test method, an easy-to-control method to induce mild tension [20,21].

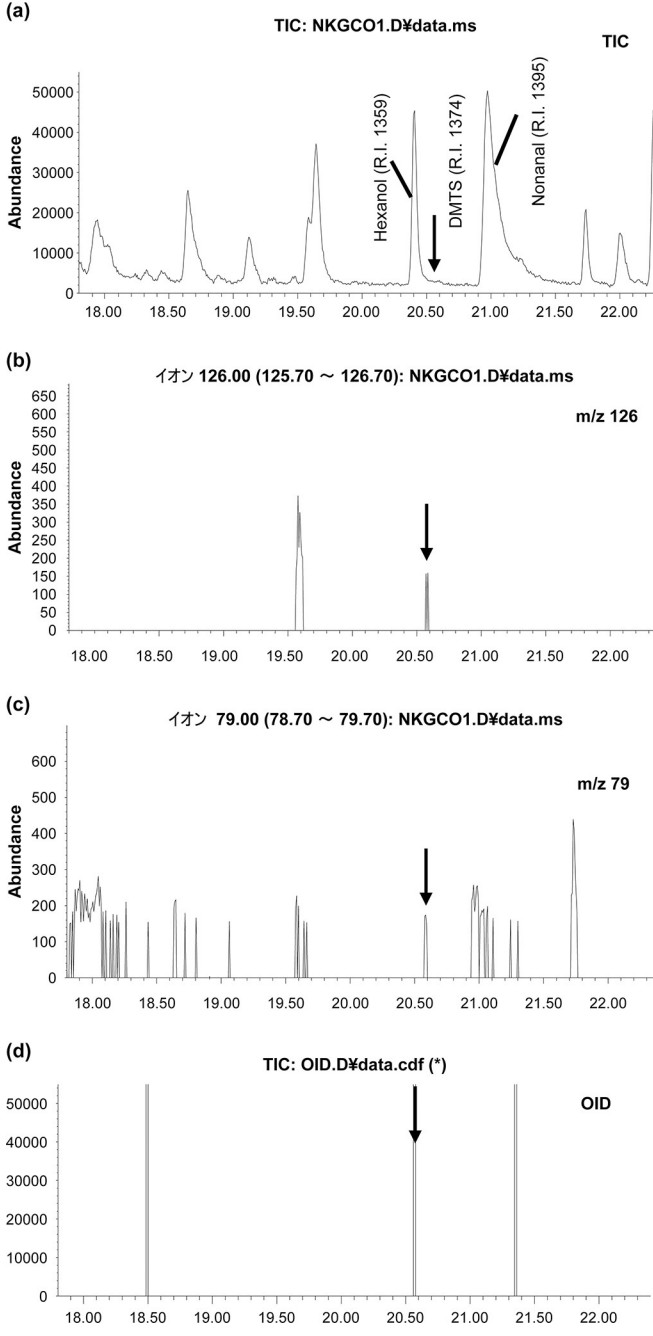

**Fig 7. Results of analysis GC-O.** A sample during a tension interview showed a peak that was close to the characteristic odor with a retention time (RT) = 20.571 minutes. Moreover, standard DMTS was detected at RT = 20.571 minutes. (a, d). At the peak of RT = 20.571 min, these ions were detected by extracting mass chromatograms with m/s = 126 and 79 as a feature of DMTS (b, c).

Collecting body odor from humans can be done in various ways, from the armpits using cotton balls [16–18], clothing [2,22], and the back and neck [23]. However, samples taken using such methods contain a mixture of bacterial metabolites of sebum and apocrine gland secretions in addition to the odor that comes directly from the body. In this study, we wanted to see the exact relationship between the changes in the body and the skin gas. For this reason,

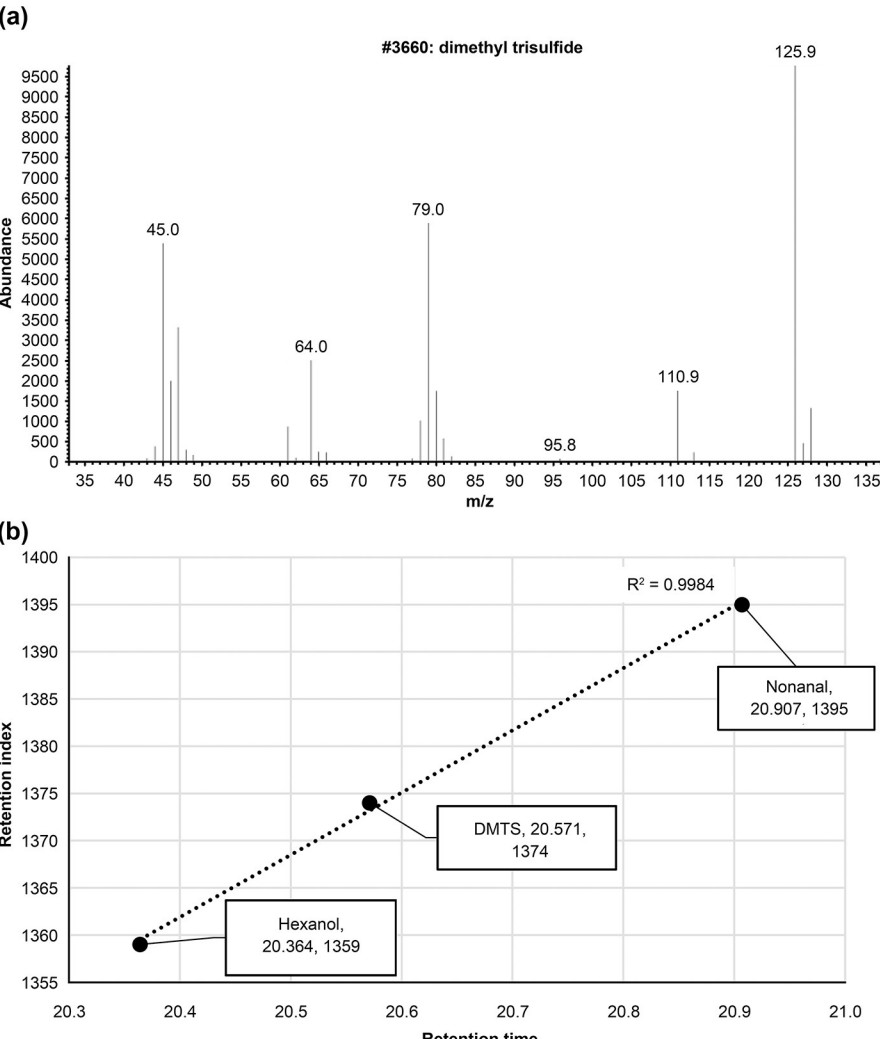

**Fig 8. Mass pattern of DMTS and correlations RT and RI.** Hexanol and nonanal were detected around the peak of RT = 0.20.571. The correlation between these RTs and the coefficient of determination (RI) of the three components registered in the MS Private Library Database of Takata Koryo Co., Ltd. was $R^2$ = 0.9984 (b).

skin-gas samples were collected from the hand because it contains fewer sebaceous glands and no apocrine glands [24].

During the tension-inducing interviews, autonomic nervous system activity was measured before, during, and after to confirm the subject's condition. The results indicated that the LF/HF values during the interviews were significantly higher than those before and after, which indicates sympathetic dominance. Salivary cortisol levels after the interview were also significantly higher than those at baseline and those while the subjects were reading were.

The test times were either morning or evening, depending on participants' availability. It was expected that the influence of the interview could be difficult to observe because of the circadian rhythm of cortisol [25,26], especially in the morning test. Therefore, control saliva was collected at the same time on another day. As a result, at the time of the interview, the stress level was higher in participants than under normal condition.

In addition, a blind sensory evaluation was conducted to ascertain any changes in odor in the skin gas caused by tension and stress. The results revealed that despite the short duration

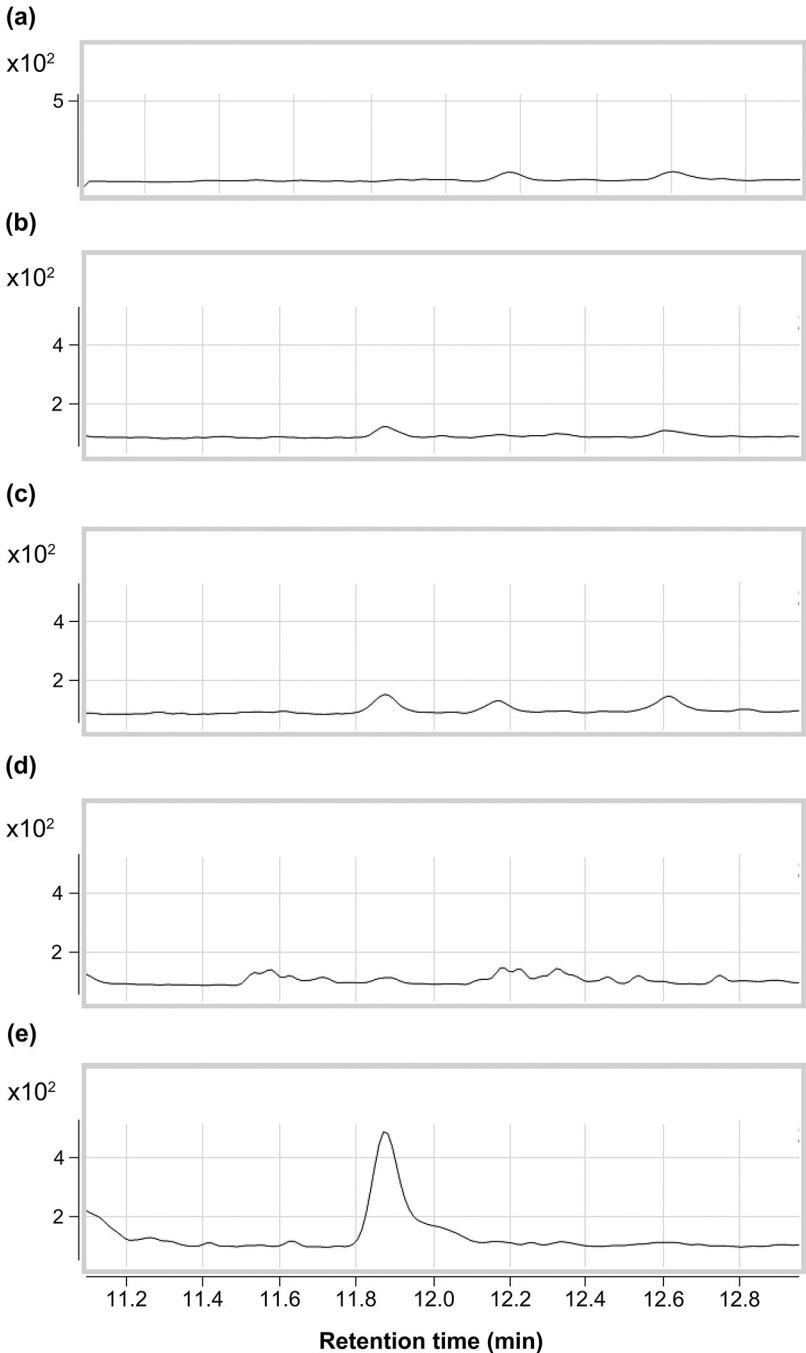

**Fig 9. Selected ion monitoring (SIM) chromatograms according to allyl mercaptan (AM) ($m/z$74).** The peak in each skin-gas sample collected during the stress interview (b, c) was detected at the same RT as that of AM, and the peak intensity of the skin-gas was higher than 2.5 ng/L and lower than 12.5 ng/L. (a): The control skin-gas without the characteristic odor representative of skin-gas with the characteristic odor; (b),(c): collected during the stress interview (panels #1 and #2), (d): Nitrogen gas containing AM (2.5 ng/L); (e): Nitrogen gas containing AM (12.5 ng/L).

(20 min), all skin-gas samples collected during the interview had a very characteristic odor, similar to stir-fried leeks. Notably, the intensity of this characteristic odor correlated positively with the rate of increase in sympathetic activity during the interviews. In another test, this phenomenon was also observed in men and nonJapanese individuals while in a state of tension.

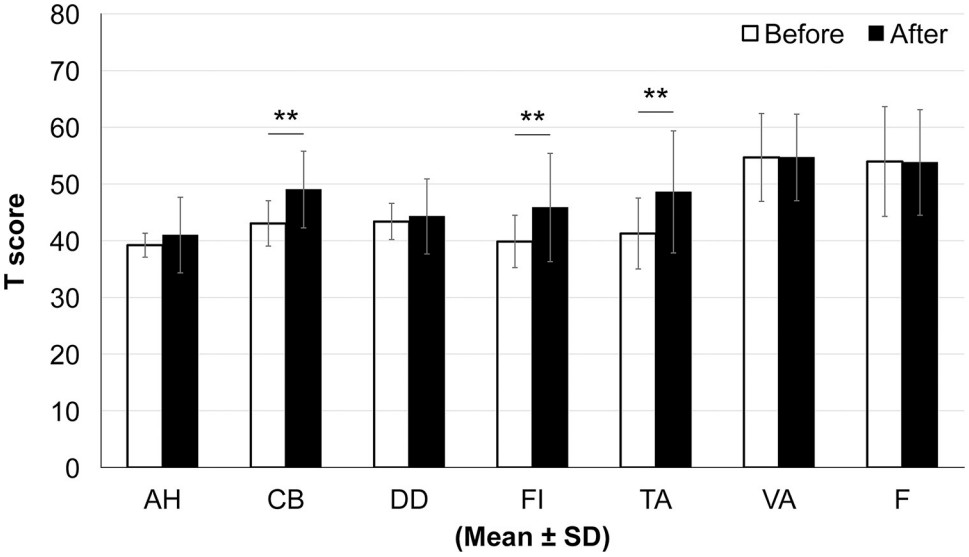

**Fig 10. Subjective effects of the tension odor.** The model tension odor, formulated according to the threshold, was continuously sniffed by participants for 2 min. POMS 2 measured the participants' subjective status at baseline and immediately after sniffing the sample. Note: AH: anger–hostility, CB: confusion–bewilderment, DD: depression–dejection, FI: fatigue–inertia, TA: tension–anxiety, VA: vigor–activity, and F: friendliness. The negative scores of "tension–anxiety" (p < 0.01), "confusion–bewilderment" (p < 0.01), and "fatigue–inertia" (p < 0.01) significantly increased after sniffing the model tension odor (Wilcoxon's signed-rank test).

These findings suggest that a characteristic odor is emitted more strongly from humans with greater emotional tension, and this phenomenon may be an instinctive human physiological reaction. In contrast, as a comparison, the results of the sensory evaluation of skin-gas samples taken during cycling exercise (>140 bpm for 20 min), meant to increase subjects' heart rate, did not identify any specific odor (Fig 6). This result suggests that a mere increase in heart rate and the accompanying increase in thermal sweating was not the direct cause of the characteristic odor released by the skin. In other words, it indicates that the odor is released as a result of psychological change.

We were able to recognize the characteristic odor through sensory evaluation. This odor can even be recognized by the general public, not just odor experts. However, the concentration of the target compound was so low that it was extremely difficult to determine it through gas chromatography. Thus, the study looked for effective sample preparation methods and analytical conditions and finally succeeded in identifying the target compounds, namely, dimethyl trisulfide and allyl mercaptan.

The characteristic odor we were seeking, emanating from the skin during stress-induced interviews, was similar to that of allyl mercaptan; however, it could not be perfectly reproduced on its own, and when the odor of dimethyl trisulfide was added, the smell was much closer to the actual smell we found that was similar to that of stir-fried leeks.

The successful identification of this component makes it possible to artificially reproduce a model odor of tension-stress. In this study, the psychological (subjective) effects of this odor were investigated: a short version of POMS2 was used to assess the subjective changes before and after 2 min of sniffing this model odor at a threshold intensity. The results revealed that smelling the model tension-stress odor increased negative emotions such as tension–anxiety (TA), fatigue–inertia (FI) and confusion-confusion (CB). However, the subjective ratings in this study were provided on a Likert-style scale, and thus, there is a possibility of an anchor effect, where the rater's memory of the previous score influences the next rating.

Nevertheless, the trends in the results suggest that odors emanating from emotionally stressed individuals may have a psychological impact on those in their vicinity.

In summary, our initial hypothesis that the psychological state of tension and stress chemically affects skin gas and its components affect human emotions has been confirmed beyond the barriers such as substance identification.

This discovery has shown several possibilities. First, the fact that the palm, which does not contain apocrine glands, releases the recognizable substances during psychological changes has opened up possibilities in biogas research. Second, this means that deodorants and fragrances can play a significant role in human communication, not etiquette alone. We are developing a practical fragrance material that does not cheat this odor with a strong scent, but harmonizes it with a weak scent and removes the odor's negative psychological effects. Third, in another area, the tension-stress odor may be used to identify a tense and stressful psychological state. Thus, this odor could be used to manage mental health conditions and facilitate smooth communication during social interactions.

Moreover, human body odor, in the most extreme cases, can provide us with clues to the location of missing victims during rescue operations or natural/manmade disasters [27,28]. In this area of rescue, the recognition of human odors induced by tension and stress may be beneficial in detecting humans in crisis.

## Conclusion

This study demonstrates that psychological changes (tension-stress) influence the components of skin gas. The main components of these substances were also identified and the effect of this odor on subjectivity was clarified. Our findings could lead to a variety of applications, including biogas, deodorants, and life-saving operations. However, several points about this phenomenon require clarification, such as the mechanism of odor generation due to psychological changes, and thus, further research is necessary.

## Supporting information

**S1 Data.**
(XLSX)

## Acknowledgments

The research was conducted through collaboration among many individuals across various fields. I am deeply grateful and indebted to all of them.

## Author Contributions

**Conceptualization:** Masako Katsuyama.

**Data curation:** Masako Katsuyama, Tomomi Narita, Masatoshi Ochiai, Naomi Kunizawa.

**Formal analysis:** Masako Katsuyama, Tomomi Narita, Naomi Kunizawa.

**Investigation:** Masako Katsuyama, Tomomi Narita, Masaya Nakashima, Kentaro Kusaba, Masatoshi Ochiai, Naomi Kunizawa.

**Methodology:** Masako Katsuyama, Tomomi Narita, Akihiro Kawaraya, Yukari Kuwahara, Masahiro Horiuchi, Koji Nakamoto.

**Writing – original draft:** Masako Katsuyama, Akihiro Kawaraya, Masahiro Horiuchi.

**Writing – review & editing:** Masako Katsuyama.

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
