## [Decision Letter · Decision Letter 0]

9 Feb 2022

PONE-D-21-36189How emotional changes affect skin odor and its impact on othersPLOS ONE

Dear Dr. KATSUYAMA,

Thank you for submitting your manuscript to PLOS ONE. After careful consideration, we feel that it has merit but does not fully meet PLOS ONE’s publication criteria as it currently stands. Therefore, we invite you to submit a revised version of the manuscript that addresses the points raised during the review process.We really appreciate for your submitting an interesting report to PLOS ONE. Both reviewers and the editor have found many interesting and important points in this report, and want to read this report in PLOS ONE. However, both reviewers have also founds some problems that should be corrected before acceptance. Many of those problems seem to be minor, but the editor estimated that you want some time to respond the reviewers' comments because of your daily businesses. You can resubmit your manuscript in anytime after you have finished the correction. We are waiting for your resubmission.

We look forward to receiving your revised manuscript.

Kind regards,

Nobuyuki Sakai, Ph.D.

Academic Editor

PLOS ONE

Journal Requirements:

Reviewers' comments:

Reviewer's Responses to Questions

**Comments to the Author**

1. Is the manuscript technically sound, and do the data support the conclusions?

Reviewer #1: Yes

Reviewer #2: Partly

2. Has the statistical analysis been performed appropriately and rigorously? 

Reviewer #1: Yes

Reviewer #2: Yes

3. Have the authors made all data underlying the findings in their manuscript fully available?

Reviewer #1: No

Reviewer #2: Yes

4. Is the manuscript presented in an intelligible fashion and written in standard English?

Reviewer #1: No

Reviewer #2: Yes

5. Review Comments to the Author

Reviewer #1: This is a very interesting paper which describes the change in skin odor under phycological stress, identification of skin gases causing the tension-stress odor and emotional effect of the odor to people by sniffing. The human skin gas has been inviting considerable attentions as a source of body odor, mosquito attractant, non-invasive medical biomarkers and so on. While a previous study showed the dermal emission of ammonia increased with physical and/or phycological stress, this study also found new skin gases causing the characteristic odor similar to sulfur-containing compounds. So, I believe this study potentially contains very valuable outcomes which merit publication. However, this manuscript has been poorly written with less objectivity. Please revise the manuscript addressing following points.

P5, L76: I do not have any criticisms on this point, but simply want to know why the study subjects were focused on nonsmoking Japanese woman? Are there any effect of smoking on the human skin gas? Why woman only?

P6, L94: I felt difficulty to catch the sentence that “Afterward, gas obtained from the sampling bags was recovered, placed in a storage bag (ANALYTIC-BARRIER™, GL Science Co., Ltd.) using connected silicone tubes.” Does this mean “Afterward, the sampling bag was connected to another storage bag using a silicon tube, and the gas in the sampling bag was transferred to the connected storage bag”? But how?

P6, L98-100: Is this the way to push the gas into another storage bag? However, the sampling bag shown in Fig.1 has only one port.

P7, L116: A reference is required for the effect of circadian rhythm on the cortisol change.

P8, L118-124: Please describe the odor assessment method more in detail to ensure the objectivity of this study. The four experts were recruited from out of author’s company (if from the company inside, please describe how to ensure the objectivity). Please demonstrate the process to characterize the odor in the bag by four experts.

P8, L132: What is the “each 1 L of skin gas”? I understand trace amount of skin gas was contained in 500 mL of nitrogen in the sampling bag. So, “skin gas” should be “gas in the sampling bag” or so. I am not sure about how to get 1 L of gas? What is “each”? The human skin gas used in this section were collected from who?

P8, L133-138: Please describe how the gas was trapped in the canister.

P8, L139-149: Who detected the gas by sniffing? One of the four experts?

P8, L150 Analysis 2: Please refer above comments on analysis 1 and revise them.

P11, L203: Please describe the model tension-stress odor more in detail. Is this a mixture of allyl mercaptan and dimethyl trisulfide? What is the solvent.

P14, L258-260: Please explore the recently published papers related to human skin gas analysis. The sulfur-containing compounds are usually found even at rest. So, I have a doubt on “none existed in the samples taken at baseline”. Please dispel my doubt by clearly showing the data obtained from the four experts, because the figure 4 seems very valuable.

P15, L276 Identifying characteristic odor: I respect author’s great efforts to identify the odor-causing gases. However, there is a big problem – control missing. I believe no peaks corresponding to allyl mercaptan and dimethyl trisulfide were found in the relax samples. So, please show the data (compare the chromatographs or so).

P16, L297-P17, L311: Based on the author’s deep experience, allyl mercaptan was successfully singled out as a causing substance. However, this section is very difficult to read. So, please sort the sentences including figure 8.

Figures captions: Please make the captions more informative. For example, which sample was used for analysis shown in Figures 6 and 8.

Reviewer #2: This study investigates relations between human subjective responses obtained by POMS and physiological responses such as gas evaporated from human skin surface of non-dominant hand, cortisol in saliva, and heart rate under a stressful state. In addition, it aims to create an odour model under stress/tense condition with which those who smell the odour are emotionally affected.

As a whole, the results in this study are shown in appropriate ways and are written in a comprehensive manner, suggesting that the main objective physiological response, skin gas, reflects subjective responses and correlates with other physiological indices reflecting sympathetic nervous system activity. Ethical consideration throughout the experiment is also appropriate. Sample size, experimental design, and analysis are properly administered as a whole. Thus this manuscript can be accepted after certain corrections and adding further explanations required to clarify the experimental procedure and its limitations. At the same time, the manuscript somewhat lacks balance of information volume in a way that some information on the experimental procedure is sufficient (such as the method to collect skin gas) and others insufficient. Especially descriptions on behavioural procedure is insufficient.

The following information includes the points that require further corrections or explanations are necessary.

Validity of the use of the term “psychological-” should be considered. The term has long been widely and inappropriately used in a wide variety of research papers to only represent human subjective responses, whereas psychology includes both subjective and objective aspects. Considering that various responses observed in animal psychology are typically non-verbal and thus must be measured only in objective manners, researchers no longer should not refer to psychological responses only as subjective. Furthermore, this study only employed POMS to evaluated participants’ subjective responses. The authors should alter the use of the term “psychological” into “subjective”, or at least define how the term is chosen to represent POMS data.

So called prescribed properties in this study should be described delicately since subjective measures should have been obtained with a more sufficient care. For example, the participants are set in the interview situation where they feel certain stress (l-232). Thus it is natural that they evaluate their subjective state in a emotionally-negative manner expecting what the experimenters anticipate. Thus how the properties are put into consideration in this study should at least be mentioned. Furthermore, the way how the experts evaluated the skin gas seems inappropriate in a sense the odours are evaluated without a blind method (l-119). If done in a blind manner, it should be noted, too.

Experimental design where the variable “stressful condition” is manipulated is not appropriate and this should also be mentioned in the manuscript. Admitting that the stressful condition is fairly controlled in a way the dependent variables are properly collected, the interview condition could have been more rigorously controlled. To be specific, a valid setting would be setting stressful (experimental condition) and non-stressful (control condition) interview/conversation conditions so that various activity levels are closer enough to each other to compare the difference in a more controlled way(l-251). Biking exercise is quantitatively and qualitatively different as an activity that it can not be rigorously compared. In that sense, cortisol level in saliva and heart rate are not compared in an ideal manner in this study. Such limitation of this study should at least be noted.

l-112

Time schedule of the experimental procudure as a whole should be illustrated in a more comprehensive way such as with a timeline figure. It is also unclear if the participants in each experiment are partly the same or different, which should also be noted.

l-113

Cortisol level can also be affected not only by the time of the day but also by the experimental surroundings which should easily affect sympathetic nervous system. Thus, the samples should have been collected on a different day at the same laboratory where the experiment was conducted. In addition, visual analogue scale (VAS) is recently used widely to evaluate various subjective responses in researches including studies on olfactory function. Furthermore, a Likert-style scale has fatal problems including anchoring effect that evaluators’s memory on the previous score affects following evaluations if done more than once in one experiment. It is partly because of the the problem that it can only be treated as a nominal or ordinal scale. Although such scales has long be used especially in the psychology field, the authors should at least explain disadvantages of such scales considering that there are accumulating numbers of studies claiming such questions.

l-193

The participants in this study are biased in age and gender. The authors should explain backgrounds of the distribution.

l-206

There are various subjective evaluation scales other than POMS. Although POMS is certainly one of the best measures and the authors properly explain the questionnaire in detail, the factors to be employed as compared with other famous measures should be simply noted. Here again, how the participants evaluated the scale (not VAS?) should be mentioned since they evaluated their state twice in the experiment.

6. PLOS authors have the option to publish the peer review history of their article (what does this mean?). If published, this will include your full peer review and any attached files.

Reviewer #1: No

Reviewer #2: No

---

## [Author Response · Author response to Decision Letter 0]

19 Apr 2022

Thank you very much for your detailed comments. I have italicized your comments and bolded my responses.

Reviewer #1: This is a very interesting paper which describes the change in skin odor under phycological stress, identification of skin gases causing the tension-stress odor and emotional effect of the odor to people by sniffing. The human skin gas has been inviting considerable attentions as a source of body odor, mosquito attractant, non-invasive medical biomarkers and so on. While a previous study showed the dermal emission of ammonia increased with physical and/or phycological stress, this study also found new skin gases causing the characteristic odor similar to sulfur-containing compounds. So, I believe this study potentially contains very valuable outcomes which merit publication. However, this manuscript has been poorly written with less objectivity. Please revise the manuscript addressing following points.

Thank you for your comments.

Below, I have provided answers and responses to some of the points you have raised.

First, the expression "the odor similar to sulfur-containing compounds " is too broad in meaning, so we changed the characteristic smell we found to a more specific expression: "the characteristic stir-fried leek-like odor".

P5, L76: I do not have any criticisms on this point, but simply want to know why the study subjects were focused on nonsmoking Japanese woman? Are there any effect of smoking on the human skin gas? Why woman only? 

<Response P5, L76>

As the skin gas sampling in this study was performed on the hands, it was necessary to avoid tobacco odor contaminants found on the hands of smokers, so the target population was non-smokers. In addition, as body odor is known to differ between men and women, the study was initially restricted to women. An explanation has been added to the text.

P6, L94: I felt difficulty to catch the sentence that “Afterward, gas obtained from the sampling bags was recovered, placed in a storage bag (ANALYTIC-BARRIER™, GL Science Co., Ltd.) using connected silicone tubes.” Does this mean “Afterward, the sampling bag was connected to another storage bag using a silicon tube, and the gas in the sampling bag was transferred to the connected storage bag”? But how? 

P6, L98-100: Is this the way to push the gas into another storage bag? However, the sampling bag shown in Fig.1 has only one port. 

<Response P6, L94 & L98-100>

I apologize for the lack of clarity in my wording.

For skin gas collection, subjects washed their hands with unscented soap and the non-dominant hand was covered with a collection bag made of polyvinyl fluoride resin film (Tedlar®) with a one-way stopcock attached. First, the stopcock was opened and excess air was pushed out of the collection bag in which the hand was placed. Next, a bag (500 mL) filled with nitrogen gas was connected via a short silicone tube and the nitrogen-filled bag was pressed on to move the nitrogen into the collection bag. The empty nitrogen bag was then removed from the stopcock. After a certain amount of time (Fig. 2), a storage bag (ANALYTIC-BARRIER™, GL Sciences Corporation) was connected via a short silicone tube to the stopcock. The collection gas was transferred from the collection bag to the storage bag by pressing on the collection bag. This gas was used as the sample.

The collection method is as described above. These operating procedures are described in the revised manuscript.

P7, L116: A reference is required for the effect of circadian rhythm on the cortisol change. 

<Response P7, L116:>

A paper on the diurnal variation of salivary cortisol has been added to the bibliography.

P8, L118-124: Please describe the odor assessment method more in detail to ensure the objectivity of this study. The four experts were recruited from out of author’s company (if from the company inside, please describe how to ensure the objectivity). Please demonstrate the process to characterize the odor in the bag by four experts.

<Response L118-124:>

The expert assessor was a person from within the company. The bags of odor were numbered and the assessors were not informed about the background of the samples. The raters shared a common perception of a distinctive, characteristic stir-fried leek-like odor and each rated the smell on a five-point scale from “no smell at all” to “strongly smelling” according to their own criteria. The operating instructions have been added.

P8, L132: What is the “each 1 L of skin gas”? I understand trace amount of skin gas was contained in 500 mL of nitrogen in the sampling bag. So, “skin gas” should be “gas in the sampling bag” or so. I am not sure about how to get 1 L of gas? What is “each”? The human skin gas used in this section were collected from who?

P8, L133-138: Please describe how the gas was trapped in the canister.

<Response P8, L132, L133-138:>

As you pointed out, firstly, the expression ' each 1 L of skin gas ' was incorrect. It has been corrected to the expression '1 L of sampled gas'. In this analysis, several skin gas samples obtained during the stress-inducing interviews were gathered by aspiration with a canister whose inner surface was deactivated and prepared with negative pressure. The operating instructions have been added.

P8, L139-149: Who detected the gas by sniffing? One of the four experts?

<Response P8, L139-149:>

In analysis 1, the odor was identified by one odor expert and one GC analyst who had learnt the odor characteristics from the odor expert.

In analysis 2, the odor was confirmed by two odor experts and a GC analyst.

This has been added to the text.

P8, L150 Analysis 2: Please refer above comments on analysis 1 and revise them.

<Response P8, L150:>

For analysis 2, we have also added a description of the odor evaluator and sample. 

P11, L203: Please describe the model tension-stress odor more in detail. Is this a mixture of allyl mercaptan and dimethyl trisulfide? What is the solvent.

<Response P11, L203:>

Allyl mercaptan (AM) and dimethyl trisulfide (DMTS) were diluted with isopropyl alcohol to create five levels of concentration (AM: DMTS = (0.5: 2.5ppb), (1: 5ppb), (2: 10ppb), (5: 25ppb), (10: 50ppb)). 10 μL of each concentration was applied to a small piece of cotton. The cotton was placed under the subjects’ noses, which they sniffed.

I added this to the revised manuscript about this process.

P14, L258-260: Please explore the recently published papers related to human skin gas analysis. The sulfur-containing compounds are usually found even at rest. So, I have a doubt on “none existed in the samples taken at baseline”. Please dispel my doubt by clearly showing the data obtained from the four experts, because the figure 4 seems very valuable.

<Response P14, L258-260:>

The expression "the odor similar to sulfur-containing compounds " is too broad in meaning, so we changed the characteristic smell we found to a more specific expression: "the characteristic stir-fried leek-like odor". The first thing that caught our attention in this study was that differences were detected by a person's sense of smell.

This is the biggest differentiation from recent reports on the analysis of body odor, where differences were found using multivariate analysis and other techniques on the GC analyzed data. Of course, we recognize that sulfur compounds are also present in normal skin gases. In this study, the most significant feature is the presence or absence of a common characteristic odor component that can be perceived by humans: specifically, the characteristic stir-fried leek-like odor, and we have adopted a method to search for the main component of this odor. A description has been added before the analytical method.

P15, L276 Identifying characteristic odor: I respect author’s great efforts to identify the odor-causing gases. However, there is a big problem – control missing. I believe no peaks corresponding to allyl mercaptan and dimethyl trisulfide were found in the relax samples. So, please show the data (compare the chromatographs or so).

<Response P15, L276:>

In the gas chromatography analysis, no peak could be detected unless it was concentrated to a high concentration, but the human sense of smell could detect the presence or absence of this characteristic odor even if it was not concentrated.

The results (Fig. 6) show that this odor was not perceived in the samples taken during relaxation and exercise.

P16, L297-P17, L311: Based on the author’s deep experience, allyl mercaptan was successfully singled out as a causing substance. However, this section is very difficult to read. So, please sort the sentences including figure 8.

Figures captions: Please make the captions more informative. For example, which sample was used for analysis shown in Figures 6 and 8.

<Response P16, L297-P17, L311:> 

The text has been reorganized and the captions amended.

Thank you very much for your detailed comments. I have italicized your comments and bolded my responses.

Reviewer #2: This study investigates relations between human subjective responses obtained by POMS and physiological responses such as gas evaporated from human skin surface of non-dominant hand, cortisol in saliva, and heart rate under a stressful state. In addition, it aims to create an odour model under stress/tense condition with which those who smell the odour are emotionally affected.

As a whole, the results in this study are shown in appropriate ways and are written in a comprehensive manner, suggesting that the main objective physiological response, skin gas, reflects subjective responses and correlates with other physiological indices reflecting sympathetic nervous system activity. Ethical consideration throughout the experiment is also appropriate. Sample size, experimental design, and analysis are properly administered as a whole. Thus this manuscript can be accepted after certain corrections and adding further explanations required to clarify the experimental procedure and its limitations. At the same time, the manuscript somewhat lacks balance of information volume in a way that some information on the experimental procedure is sufficient (such as the method to collect skin gas) and others insufficient. Especially descriptions on behavioural procedure is insufficient.

The following information includes the points that require further corrections or explanations are necessary.

Thank you for your comments. I apologize for the lack of organization in the text. I will provide an overall organized explanation before responding to each comment. We have also reorganized the abstract. First, the expression "the odor similar to sulfur-containing compounds " is too broad in meaning, so we changed the characteristic smell we found to a more specific expression: "the characteristic stir-fried leek-like odor".

The study consists of three points.

#1 The discovery of an odor like that of stir-fried leeks from the non-dominant hand of subjects in a state of stress, as supported by physiological indices

#2 Identification of the components of the characteristic leek odor

#3 By smelling the identified components, even in non-stressful situations, one can subjectively perceive tension and fatigue. In other words, it was shown that feelings can be propagated by smell.

Below, I have provided answers and responses to some of the points you raised.

Validity of the use of the term “psychological-” should be considered. The term has long been widely and inappropriately used in a wide variety of research papers to only represent human subjective responses, whereas psychology includes both subjective and objective aspects. Considering that various responses observed in animal psychology are typically non-verbal and thus must be measured only in objective manners, researchers no longer should not refer to psychological responses only as subjective. Furthermore, this study only employed POMS to evaluated participants’ subjective responses. The authors should alter the use of the term “psychological” into “subjective”, or at least define how the term is chosen to represent POMS data.

<Response :>

I understand your point and have changed 'psychological' to 'subjective'.

l-232

So called prescribed properties in this study should be described delicately since subjective measures should have been obtained with a more sufficient care. For example, the participants are set in the interview situation where they feel certain stress (l-232). Thus it is natural that they evaluate their subjective state in a emotionally-negative manner expecting what the experimenters anticipate. Thus how the properties are put into consideration in this study should at least be mentioned. 

<Response: L-232> 

The interview was, as you stated, a situation where we expected the participants to feel stressed. We showed that the interview produced a stress state, as we had expected, by checking fluctuations in the physiological indicators via electrocardiographic activity and salivary cortisol levels.

l-119

Furthermore, the way how the experts evaluated the skin gas seems inappropriate in a sense the odors are evaluated without a blind method (l-119). If done in a blind manner, it should be noted, too.

<Response: L-119>

The skin gas samples were numbered and the assessors were not informed of the background of the samples to help ensure a blind evaluation. The raters shared the perception of a distinctive odor like that of stir-fried leeks and rated its intensity on a five-point scale from “not smelling at all” to “strongly smelling”, according to their respective evaluation criteria. I have added the operating instructions. 

Experimental design where the variable “stressful condition” is manipulated is not appropriate and this should also be mentioned in the manuscript. Admitting that the stressful condition is fairly controlled in a way the dependent variables are properly collected, the interview condition could have been more rigorously controlled. To be specific, a valid setting would be setting stressful (experimental condition) and non-stressful (control condition) interview/conversation conditions so that various activity levels are closer enough to each other to compare the difference in a more controlled way(l-251). Biking exercise is quantitatively and qualitatively different as an activity that it can not be rigorously compared. In that sense, cortisol level in saliva and heart rate are not compared in an ideal manner in this study. Such limitation of this study should at least be noted.

<Response :> 

Thank you very much for the advice. In this exam, we created an environment where subjects were interviewed by people they did not know as a means of creating stress. Regardless of the contents of the interview, talking with a stranger can be stressful. For the relaxed state, we asked the subjects to read magazines that didn’t contain any stimulating topics.

We needed to prove that the characteristic leek-like odor produced during the interview was not simply due to increased sweating. Therefore, we had subjects exercise on a stationary bike as a means of increasing their heart rate and perspiration without increasing stress levels, and checked the odor of skin gases collected during this exercise.

l-112

Time schedule of the experimental procudure as a whole should be illustrated in a more comprehensive way such as with a timeline figure. It is also unclear if the participants in each experiment are partly the same or different, which should also be noted.

<Response l-112:> 

A time schedule for the entire skin gas sampling experiment process during tension-induced interviews has been added. 

Participants in the verification of the effect for subjective reaction of the model tension odor were some of the participants in the interview test and were asked to cooperate on another day.

l-113

Cortisol level can also be affected not only by the time of the day but also by the experimental surroundings which should easily affect sympathetic nervous system. Thus, the samples should have been collected on a different day at the same laboratory where the experiment was conducted. In addition, visual analogue scale (VAS) is recently used widely to evaluate various subjective responses in researches including studies on olfactory function. Furthermore, a Likert-style scale has fatal problems including anchoring effect that evaluators’s memory on the previous score affects following evaluations if done more than once in one experiment. It is partly because of the the problem that it can only be treated as a nominal or ordinal scale. Although such scales has long be used especially in the psychology field, the authors should at least explain disadvantages of such scales considering that there are accumulating numbers of studies claiming such questions.

<Response l-113:> 

I understand your point about the timing of the saliva collection.

In this experiment, although the saliva from after the subjects read was collected during the same time of day the saliva was collected at home, the results were also included in the graph (fig. 4).In addition, I understand that the POMS Likert scale references the previous score, which causes an anchoring effect on the next evaluation. In the main text, we have added an explanation of its disadvantages.

l-193

The participants in this study are biased in age and gender. The authors should explain backgrounds of the distribution.

<Response l-193:>

As the skin gas sampling in this study was performed on the hands, it was necessary to avoid tobacco odor contaminants found on the hands of smokers, so the target population was non-smokers. In addition, as body odor is known to differ between men and women and with age, the study was initially restricted to women. I have added an explanation about this to the text.

l-206

There are various subjective evaluation scales other than POMS. Although POMS is certainly one of the best measures and the authors properly explain the questionnaire in detail, the factors to be employed as compared with other famous measures should be simply noted. Here again, how the participants evaluated the scale (not VAS?) should be mentioned since they evaluated their state twice in the experiment. <Response l-206:> 

As you pointed out, there are various subjective evaluation scales. POMS was selected because the subjects of this study were healthy people and POMS has the ability to cover seven subjective emotional states. Subjects evaluate themselves twice in one experiment on a 5-point Likert scale. We described the disadvantages of answering using the Likert scale within the same study.

---

## [Decision Letter · Decision Letter 1]

12 Jun 2022

How emotional changes affect skin odor and its impact on others

PONE-D-21-36189R1

Dear Dr. KATSUYAMA,

Thank you for your interesting manuscript. I have decided accepting your manuscript as an article in Plos One, but a reviewer had commented on your revision. You do not need to answer this comment any more, but the comment should be help your study in the future. Please keep the comment described below.

We’re pleased to inform you that your manuscript has been judged scientifically suitable for publication and will be formally accepted for publication once it meets all outstanding technical requirements.

Kind regards,

Nobuyuki Sakai, Ph.D.

Academic Editor

PLOS ONE

Additional Editor Comments (optional):

Reviewers' comments:

Reviewer's Responses to Questions

**Comments to the Author**

1. If the authors have adequately addressed your comments raised in a previous round of review and you feel that this manuscript is now acceptable for publication, you may indicate that here to bypass the “Comments to the Author” section, enter your conflict of interest statement in the “Confidential to Editor” section, and submit your "Accept" recommendation.

Reviewer #1: All comments have been addressed

Reviewer #2: All comments have been addressed

2. Is the manuscript technically sound, and do the data support the conclusions?

Reviewer #1: Yes

Reviewer #2: Yes

3. Has the statistical analysis been performed appropriately and rigorously? 

Reviewer #1: Yes

Reviewer #2: Yes

4. Have the authors made all data underlying the findings in their manuscript fully available?

Reviewer #1: Yes

Reviewer #2: Yes

5. Is the manuscript presented in an intelligible fashion and written in standard English?

Reviewer #1: Yes

Reviewer #2: Yes

6. Review Comments to the Author

Reviewer #1: Thank you very much for your great efforts to improve the manuscript. I think it is now acceptable for publication.

Reviewer #2: The authors have appropriately corrected the whole content, reflecting my suggestions. Thus no major revision will be necessary. The paper can be accepted with minor corrections. Yet the experimental design should be discussed a little further in the manuscript regarding its limitation in condition control since the authors have not fully responded to my suggestions (see below). I understood the authors's explanations but I believe they have not answered to my questions fully. How to control independent variables in a rigorous manner should be reconsidered closely again.

ref.

Experimental design where the variable “stressful condition” is manipulated is not appropriate and this should also be mentioned in the manuscript. Admitting that the stressful condition is fairly controlled in a way the dependent variables are properly collected, the interview condition could have been more rigorously controlled. To be specific, a valid setting would be setting stressful (experimental condition) and non-stressful (control condition) interview/conversation conditions so that various activity levels are closer enough to each other to compare the difference in a more controlled way(l-251). Biking exercise is quantitatively and qualitatively different as an activity that it can not be rigorously compared. In that sense, cortisol level in saliva and heart rate are not compared in an ideal manner in this study. Such limitation of this study should at least be noted.

<response>

Thank you very much for the advice. In this exam, we created an environment where subjects were interviewed by people they did not know as a means of creating stress. Regardless of the contents of the interview, talking with a stranger can be stressful. For the relaxed state, we asked the subjects to read magazines that didn’t contain any stimulating topics. We needed to prove that the characteristic leek-like odor produced during the interview was not simply due to increased sweating. Therefore, we had subjects exercise on a stationary bike as a means of increasing their heart rate and perspiration without increasing stress levels, and checked the odor of skin gases collected during this exercise.</response>

7. PLOS authors have the option to publish the peer review history of their article (what does this mean?). If published, this will include your full peer review and any attached files.

Reviewer #1: No

Reviewer #2: No

---

## [Editor Report · Acceptance letter]

22 Jun 2022

PONE-D-21-36189R1 

How emotional changes affect skin odor and its impact on others

Dear Dr. Katsuyama:

I'm pleased to inform you that your manuscript has been deemed suitable for publication in PLOS ONE. Congratulations! Your manuscript is now with our production department. 

Kind regards, 

on behalf of

Dr. Nobuyuki Sakai 

Academic Editor

PLOS ONE